ecology

allometry, colony, hive, scaling, bumblebee

**Author for correspondence:**
C. D. Perl
e-mail: craig.perl@zoologi.su.se

[†]Present address: School of Life Sciences, Arizona State University, Tempe, AZ, USA.
[‡]Present address: Department of Physics, University of Jyvaskyla, 40014 Jyvaskyla, Finland.

# Substantial variability in morphological scaling among bumblebee colonies

C. D. Perl[1,2,†], Z. B. Johansen[1], V. W. Jie[1],
Z. Moradinour[1], M. Guiraud[1], C. E. Restrepo[1],
A. Miettinen[3,4,‡] and E. Baird[1,2]

[1]Department of Zoology, Stockholm University, Stockholm 106 91, Sweden
[2]Department of Biology, Lund University, Lund 223 62, Sweden
[3]Swiss Light Source, Paul Scherrer Institute, 5234 Villigen, Switzerland
[4]Institute for Biomedical Engineering, University and ETH Zurich, 8092 Zurich, Switzerland

 CDP, 0000-0002-9911-1207; AM, 0000-0003-3132-0544;
EB, 0000-0003-3625-3897

Differences in organ scaling among individuals may play an important role in determining behavioural variation. In social insects, there are well-documented intraspecific differences in colony behaviour, but the extent that organ scaling differs within and between colonies remains unclear. Using 12 different colonies of the bumblebee *Bombus terrestris*, we aim to address this knowledge gap by measuring the scaling relationships between three different organs (compound eyes, wings and antennae) and body size in workers. Though colonies were exposed to different rearing temperatures, this environmental variability did not explain the differences of the scaling relationships. Two colonies had differences in wing versus antenna slopes, three colonies showed differences in wing versus eye slopes and a single colony has differences between eye versus antenna slopes. There are also differences in antennae scaling slopes between three different colonies, and we present evidence for putative trade-offs in morphological investment. We discuss the utility of having variable scaling among colonies and the implication for understanding variability in colony fitness and behaviour.

## 1. Introduction

The relative size of an organ with respect to body size impacts not only on physical and physiological regulation but also on how individuals perceive and interact with their world [1,2]. The relationship between organ size and body size can be isometric (where organ size scales proportionately with body size) or allometric (where organ size scales disproportionately with

body size) and the scaling can be positive, in which organs are relatively larger per unit body size, or negative, in which they are relatively smaller per unit body size.

Unlike for most animals, the size of organs relative to body size in the workers of eusocial insects like bumblebees impacts colony, rather than individual, fitness [3]. Workers in a eusocial insect colony are responsible for foraging and brood care, and any failure to perform these behaviours affects the capacity of the colony to produce a new generation of queens and drones. Therefore, it is ultimately worker behaviour, as determined by physiology and morphology, that determines colony fitness.

Foraging, locomotion and particularly sensory capability are determined by organ size [4–8] and differences in allometric scaling will change the relative size of organs. This can have a profound impact on worker behaviour and therefore colony performance and fitness. Variation in organ size can predict flower usage [9] and foraging capability [10] in bumblebees. Stingless bees have particularly steep eye scaling slopes, providing relatively larger eyes that permit foraging in dimmer light compared with bees of equal body sizes that have shallower scaling slopes [5]. How different organs scale with respect to body size will affect individual worker performance which ultimately determines the success of the colony.

Despite the importance of worker organ scaling for colony performance, very few investigations have explored intraspecific allometric differences among social insect colonies. Owen & Harder [3] examined scaling among different colonies of *Bombus huntii* and *B. occidentalis*. They found significant intercept differences among colonies when regressing proboscis length against wing length, but no differences among slopes [3]. They concluded that these differences are heritable and may influence colony fitness [3]. Similarly, there are significant genetic-based inter-colony differences in scape allometry among Australian sugar ant (*Camponotus consobrinus*) soldiers [11] but no differences in head width allometry among colonies of *C. novaeboracensis* [12]. There were also no allometric differences found among colonies of fire ants (*Solenopsis invicta*) [13], though the authors speculate that a lack of differences may be due to the low number of colonies sampled (four). Nonetheless, analysis on a similar number of wood ant (*Formica rufa*) colonies showed that eye allometry varied considerably [14]. As well as morphological variability among colonies, eusocial Hymenopteran colonies are also known to exhibit significant behavioural variability among workers. Colony differences exist in foraging [10,15], learning [16], the degree of worker specialization [17] and aggression [18,19] but the extent to which this is related to variation in morphology and scaling relationships between colonies remains unknown.

To begin to address this, we examined the scaling relationship of three different organs in workers from 12 *B. terrestris* colonies. We selected organs that are of high functional significance: wing size determines flight capacity through wing beat frequency [20], eye size determines visual acuity, sensitivity and resolution [21–23] and antennal length determines olfactory sensitivity [7]. These three organs represent both sensory (eye, antenna) and non-sensory (wing) functions but together contribute to foraging efficiency by affecting locomotion [20], navigation and feeding choices [9,21,24].

## 2. Methods

### 2.1. Animals

The investigation was conducted on commercial *Bombus terrestris* (L., 1758) colonies bought from BioBest (Westerlo, Belgium) in November 2019 (colonies one to four) and Koppert (Berkel en Rodenrijs, The Netherlands) in January 2020 (colonies five to 12). Colonies were reared in the dark at either 23°C or 32°C in climate-controlled cabinets (Panasonic MIR, 123 L) as part of another experiment. As part of that experiment, colonies one through four were split into two, with one half being reared at 23°C and the other half at 32°C (electronic supplementary material, table S1). Colonies five through 12 were not split; whole colonies were reared at either 23°C or 32°C (electronic supplementary material, table S1). Any effect of temperature was controlled for in the statistical analysis. The colonies were fed *ad libitum* with 50% sugar water solution and fresh-frozen, organic pollen every 2–3 days (Naturprodukter, Raspowder Bipollen). All existing bees in each colony were marked after seven days with non-toxic paint (Färgpenna, Lackstift, Biltema), an additional 7–16 days were allowed to elapse, and then newly emerged individuals were marked with individual number tags. A total of 121 *B. terrestris* workers contributed morphological data, but contributions were unequal among colonies and organs (electronic supplementary material, figure S1) due to natural differences in survival and accidental destruction of some samples during collection.

## 2.2. Morphological measurements

Bumblebees were euthanized using ethyl acetate and their mass was documented within 5 min of death followed by dissections. Antennae were dissected from bumblebee workers, laid flat on 1 mm grid paper and photographed using a Nikon D810 with a Nikkor 60 mm micro-lens. One antenna from each individual was selected at random and its length measured in ImageJ [25]. Right forewings were dissected from bumblebee workers and photographed using the camera set-up as described above. Wing size was measured as centroid size, defined as the square root of the sum of squared distances between all landmarks and their centroid. Wing landmarks followed Gerard *et al.* [26]. Landmark coordinates were marked using ImageJ [25] and centroid size was calculated using the geomorph package [27] in R v. 3.5.1 [28]. Body size was measured as fresh mass to the nearest microgram using a balance (BP 310S, Sartorius).

## 2.3. Eye surface area micro-CT scanning and reconstruction

Eye size was determined from eye surface measurements acquired from three-dimensional volume rendering of the bumblebee heads acquired from X-ray microtomographic (micro-CT) analyses [29,30]. To improve contrast in the X-ray images, the heads were stained using phosphotungstic acid (PTA). A small incision into the top of the head (to facilitate chemical penetration) was followed by decapitation and placing the sample directly into 70% ethanol + 0.5 g PTA [31] for 10 days.

The micro-CT scanning was conducted at the TOMCAT beamline of the Swiss Light Source (beamtime numbers: 20191425, 20190641). The X-ray beam was monochromatic with 15–20 keV average energy and 100 mm propagation distance for distinct phase contrast. A magnification of 4.0× (colonies one to four) or 2.0× (colonies 5–12) and exposure time of 60 ms was used (the difference in magnifications was due to differences in the set-up used during each beamtime and did not affect the precision of the subsequent measurements). For each sample, 2001 projection images of 2560 × 2048 pixels each were acquired, with 3.25 μm pixel size, over 180° of sample rotation. The projection images were processed with the Paganin phase retrieval method [32] and reconstructed into three-dimensional images with the gridrec algorithm [33]. Ring removal was done with sarepy sorting or sarepy all [34]. Scan slices were first cropped using Drishti 2.6.4 [35] and imported into Amira 6.2.0 (ThermoFisher Scientific). A digital reconstruction of the surface area was calculated and its surface area determined.

## 2.4. Statistics

All statistics were conducted using R v. 3.5.1 [28]; mixed-effects models were fitted using the lme4 package [36] and type III ANOVA tables were generated using lmerTest [37]. *Post hoc* pairwise comparisons and coefficient estimates were generated using the emmeans package [38]: comparisons of intercept was implemented using the function 'emmeans' and comparison of slopes was implemented using the function 'emtrends' [38]. Degrees of freedom for pairwise comparisons were estimated using the Kenward-Rogers method and *p*-values were Tukey-adjusted [38,39]. Differences in mean body mass among colonies were calculated using an ANOVA followed by *post hoc* pairwise comparisons using Tukey's honestly significant difference test.

Factor analysis and subsequent cluster analysis was conducted using the FactoMineR package [40]. Due to the inclusion of colony affiliation as a factor, factor analysis of mixed data (FAMD) was employed over principal components analyses, allowing the inclusion of categorical factors [40]. After dimension reduction, hierarchical cluster analysis was implemented using the function HCPC [40,41]. The FAMD and the subsequent cluster analysis, respectively, inform which factors are most important for explaining the variability in our data and which colonies are most similar. The predicted values of the allometric slopes for each organ along with colony affiliation were the data used for the FAMD. Colonies that cluster together are likely to have similar allometric slopes, given that they are occupying a similar position along with the FAMD axes.

## 2.5. Model selection and factor estimations

Likelihood ratio tests (LRT) are often used to compare models fitted under maximum likelihood. Once the minimum adequate model had been ascertained, *post hoc* comparisons are made with the same model structure but using restricted maximum likelihood (REML). However, it has been noted that

LRT may be somewhat anti-conservative when estimating mixed-effects models, especially with small sample sizes [39]. Using the lmerTest package [37], we fit models using REML and then estimated the significance of the fixed effects using $p$-values generated via an ANOVA with Satterthwaite's approximations. This approach produces more appropriate type 1 error rates, but does potentially cause a loss of power [39]. Using the step function to test differences in AIC yielded the same significant fixed effects as Satterthwaite's approximation, but given the reduction in type 1 error, we present the more conservative Satterthwaite's approximation.

Satterthwaite's approximation indicated that the initially fitted maximal model with a three-way interaction term was appropriate to explain our data (electronic supplementary material, table S2). $Log_{10}$(organ size) was modelled as a response to a three-way interaction between $log_{10}$(mass), colony affiliation and organ type. We also fitted a random intercept of individual nested within temperature. This accounted for the different temperature treatments, the repeated measures taken from individual bees and the structure caused by bees being exposed to only a single-temperature treatment. The model structure was as such:

$$\log_{10}(\text{organ size}) \sim \log_{10}(\text{mass}) * \text{Colony} * \text{Organ} + (1|\text{Temperature/Individual}).$$

# 3. Results

Slope, intercept and colony comparisons that are not explicitly mentioned in the results section were non-significantly different from each other and have been omitted for the sake of brevity. Nearly all colonies had non-significantly different mean fresh body masses, aside from colonies two and eleven (electronic supplementary material, figure S2). Worker mean fresh body mass was higher in colony two than most other colonies ($p < 0.001$), but non-significantly different from colony 11 (electronic supplementary material, figure S2). Colony 11 had a higher mean fresh body mass than most other colonies ($p < 0.01$), but was non-significantly different from colonies two, four and 12 (electronic supplementary material, figure S2).

## 3.1. Mixed-effects model

### 3.1.1. Organ size independent of organ identity

In general, organ size increased with log mass ($F = 155.00_{121,1}$, $p < 0.001$), and when averaged across all organs, there was no significant difference in organ size between colonies ($F = 0.62_{121,11}$, $p = 0.81$). All scaling relationships have a slope less than 1 and therefore have negative allometry, meaning smaller individuals have proportionally larger organs.

### 3.1.2. Mean organ size (intercept) across colonies

There was a single example of a difference in mean size of the same organ across colonies. The mean antenna size (i.e. intercept) in colony eight was higher than the mean antenna size in colony three ($t = 4.13_{121,162}$, $p < 0.03$), indicating that for a given body size, workers in colony eight will have larger antennae than in colony three.

### 3.1.3. Organ slopes independent of colony affiliation

Log mass and organ also had a significant two-way interaction ($F = 9.22_{121,2}$, $p < 0.001$), indicating that different organs have different allometric slopes (figure 1). Wing size scales with a steeper slope than antenna size ($t = 3.29_{121,144}$, $p < 0.01$) and eye size ($t = 2.85_{121,131}$, $p < 0.001$). There was no significant difference between the allometric slopes of eye size and antenna size ($t = 0.68_{121,146}$, $p = 0.78$). This means that as body size increases, wing size gets proportionally larger than both eyes and antennae.

### 3.1.4. Comparing slopes of organs within colonies

Within colony three, wing size scaled with a steeper slope than antenna length ($t = 5.17_{121,146}$, $p < 0.001$), indicating that as workers from colony three get bigger, their wings get proportionally larger than their antennae (figures 2 and 3). Within colony seven, wing size scaled with a steeper slope than eye area ($t = 2.47_{121,136}$, $p < 0.04$), indicating that as workers from colony seven get bigger, their wings get

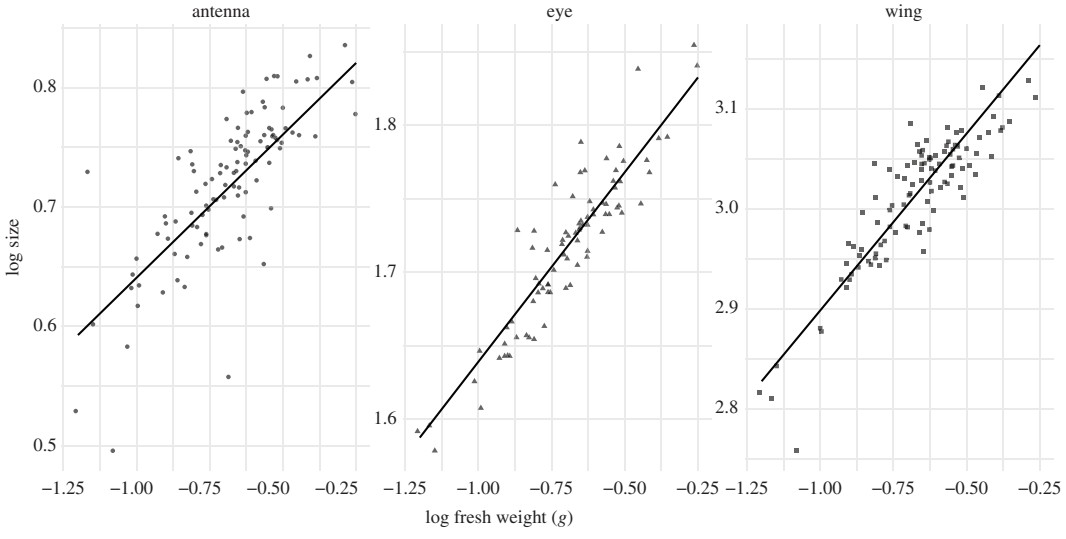

**Figure 1.** Allometric scaling slopes for wing, eye and antenna size averaged across colonies.

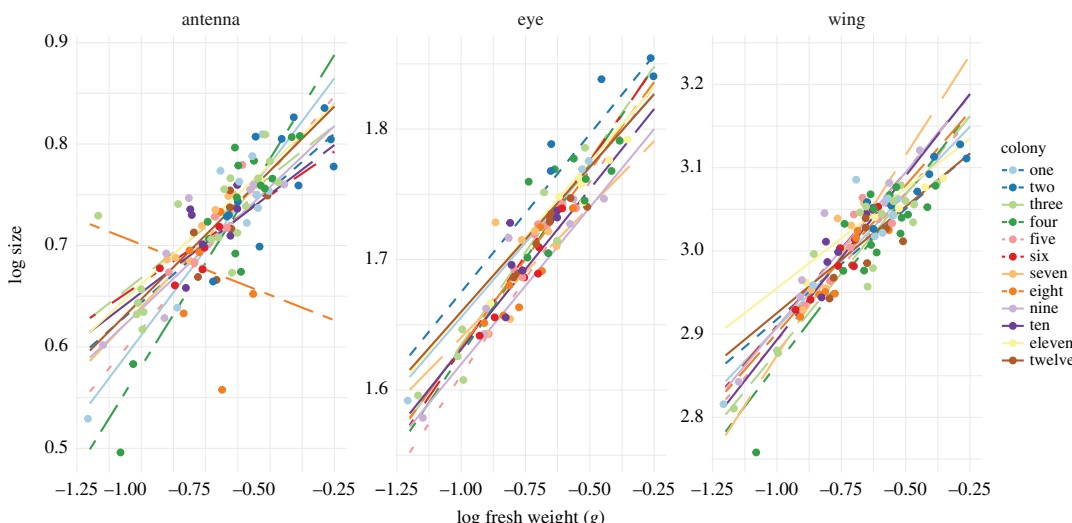

**Figure 2.** Allometric scaling relationships for wing, eye and antenna size as a function of body mass for all 12 colonies.

proportionally larger than their eyes (figures 2 and 3). Within colony eight, wing size ($t = 3.13_{121,134}$, $p < 0.01$) and eye area ($t = 3.83_{121,134}$, $p < 0.01$) scaled with a steeper slope than antenna length (figures 2 and 3). Therefore, as workers from colony eight increase in size, their wings and eyes get proportionally larger than their antennae. Within colony nine, wing size scaled with a steeper slope than antenna length ($t = 2.82_{121,126}$, $p < 0.02$) and eye area ($t = 2.82_{121,126}$, $p < 0.02$), indicating that as workers from colony nine increase in size, their wings will get proportionally larger than their antennae (figures 2 and 3).

### 3.1.5. Comparing slopes of organs across colonies

There was a significant three-way interaction between log organ size, log mass and colony affiliation ($F_{121,134} = 1.87$, $p = 0.02$), indicating that organs within and between different colonies have different allometric slopes (figures 2 and 3). Pairwise comparisons revealed the slope of antenna scaling in colony four was steeper than the antenna scaling slopes in colony three ($t = 4.19_{121,169}$, $p = 0.02$) and colony eight ($t = 4.04_{121,199}$, $p = 0.03$), meaning that antennae size increases faster with body size in colony four compared with colonies three and eight (figures 2 and 3). Wing size in colony four also scaled with a steeper slope than antenna length in colony eight ($t = 3.96_{121,202}$, $p = 0.04$), meaning that wing size increases faster with body size in colony four compared with colony eight (figures 2 and 3).

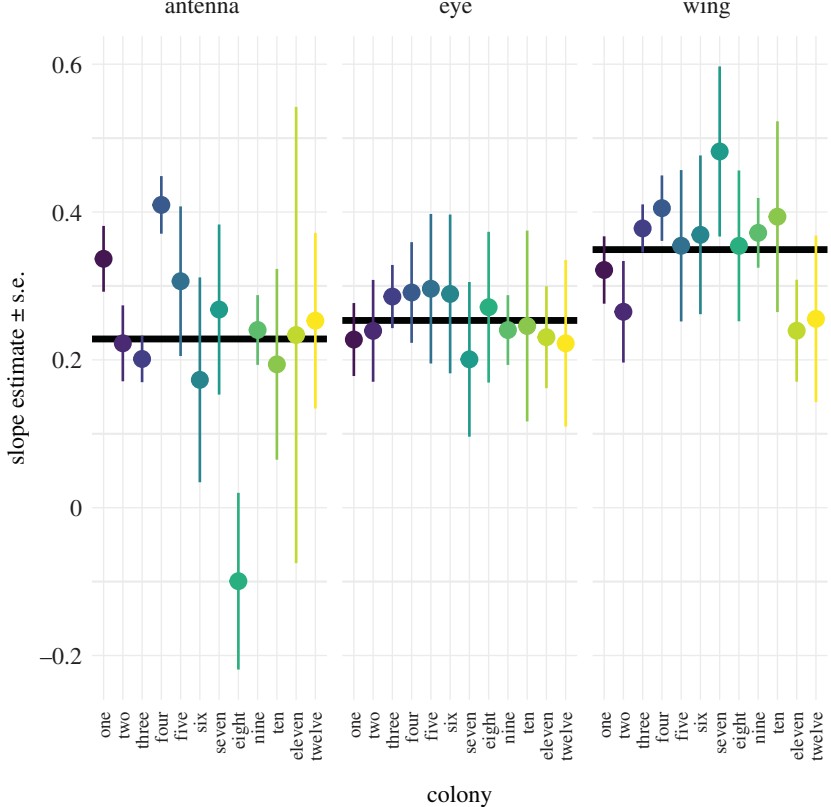

**Figure 3.** Slopes (or scaling exponents) estimates for allometric scaling relationships for wing, eye and antenna size as a function of body mass for all 12 colonies with standard errors.

**Table 1.** Variance explained by first five dimensions from FAMD.

| dimension | 1 | 2 | 3 | 4 | 5 |
|---|---|---|---|---|---|
| variance | 3.87 | 1.04 | 1.02 | 1.00 | 1.00 |
| % of variance | 25.78 | 6.90 | 6.78 | 6.70 | 6.67 |
| cumulative % of var. | 25.78 | 32.68 | 39.47 | 46.16 | 52.83 |

## 3.2. Factorial analysis of mixed data

We used FAMD and hierarchical clustering to assess which colonies had similar allometric slopes. Estimates of the allometric slopes from each organ from every colony, along with colony affiliation were the raw data used for the FAMD. Contributions to the first principal axis were mostly due to differences in colony affiliation (46%) and the eye scaling slope (30%) (table 1). The second principal axis was also strongly influenced by colony affiliation (49%) and antenna scaling slope (38%). The contribution of wing size scaling slope is more evenly divided across the first three principal axes (table 1). Hierarchical clustering revealed six clusters (figure 4), colonies two, 11 and 12 formed one cluster, colonies one, nine and 10 a second cluster and a third cluster was composed of colonies three, five and six. Colonies four, seven and eight formed separate, isolated clusters (figure 4). Across all three organs, some colonies have slopes more alike each other than to other colonies. Therefore, there is a difference in how colonies behave allometrically.

## 4. Discussion

We investigated the prevalence of allometric differences among 12 bumblebee colonies and found substantial variation in how eyes, wings and antennae scale with body size. We found that there are significant allometric differences in organ scaling among workers from different colonies. Workers from a single colony showed

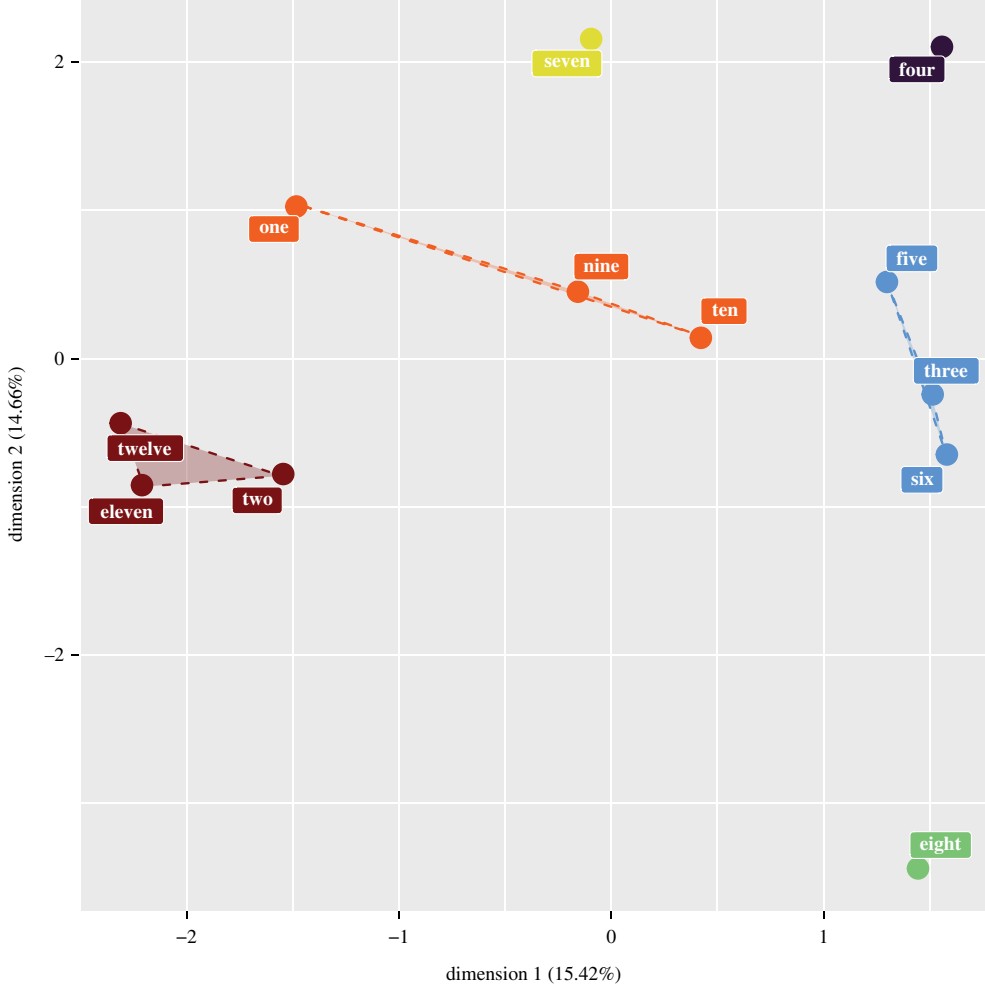

**Figure 4.** Nest clusters after FAMD performed on the slopes of scaling relationships of antenna, eyes and wings within 12 different bumblebee colonies.

differences in slope between all three organs and two colonies showed differences in slope among two of their organs. There were multiple examples of the same organ scaling with different slope between different colonies and a single example of a difference in the intercept of antenna length allometry between two colonies.

Allometric differences in organ scaling, such as the ones observed in this study, likely contribute to observed variation in colony-level behaviour and fitness. We have shown that the allometric scaling rules for antennae, wings and eyes are not fixed within *B. terrestris* but can show substantial variability among colonies. These differences are not always captured when examining scaling relationships across the entire species or population; eye and antenna scaling slopes were the same when averaged across all colonies. If colony affiliation is ignored, wing scaling slopes differed from the slopes of the two sensory structures. Allometric intercepts seem to follow more consistent patterns, with only a single difference in mean antenna size. All three organs scaled with negative allometry, and this is consistent with data from many other investigations into insect allometry [4,42–44].

FAMD and hierarchical clustering reveals that some colonies have slopes that are more similar than others (figure 4). Nest affiliation and the antenna scaling slope are the most influential factors that differentiate colonies from each other. The clustering of different colonies reflects the differences in scaling slope of various organs. Of the colonies that have different scaling slopes for some of their organs (three, four, seven, eight and nine), three of those are isolated and do not cluster with other colonies (four, seven and eight).

## 4.1. Allometry among colonies

Our data show that allometric scaling rules differ between colonies. This is consistent with previous investigations into inter-colony scaling in bumblebees [3] and wood ants [14]. Our data show that the relative importance of a given organ with increasing worker size is different among bumblebee

**Table 2.** Percentage contributions of variables to FAMD dimensions.

| dimension | 1 | 2 | 3 | 4 | 5 |
|---|---|---|---|---|---|
| antenna slope | 2.23 | 38.03 | 9.57 | <0.01 | <0.01 |
| eye slope | 29.50 | 1.57 | 18.51 | <0.01 | <0.01 |
| wing slope | 21.93 | 11.70 | 16.02 | <0.01 | <0.01 |
| colony | 46.33 | 48.71 | 55.90 | >99.99 | >99.99 |

colonies. The FAMD indicates that there is a potential trade-off in organ investment (table 2), with the first principal axis having large contributions from wing size and eye size slope, but minimal contribution from antenna size slope, which mostly contributes to the second principal axis. This implies that colonies that steepen the slope of their antenna allometry may not be able to do the same for the other organs, thereby suggesting a putative trade-off in resource allocation [45].

Our data show that both intercepts and slopes differ among colonies, which contrasts with Owen & Harder [3], who found only differences in intercept. However, the many differences between these studies make the findings difficult to compare; Owen & Harder [3] used wing size as a proxy for body size and only compared tongue lengths in wild colonies of *B. huntii* and *B. occidentalis*, while we used body mass as a proxy of size and measured antennae, eyes and wings of laboratory-reared *B. terrestris*. Nonetheless, both studies find substantial allometric variability among colonies.

## 4.2. Implications for worker behaviour and colony fitness

The laboratory-reared history of these colonies, with their presumed lack of selective pressure, makes the role of selection in generating these differential scaling relationships somewhat tenuous. In a field setting, having a steeper eye scaling slope would imply that a larger eye (with better vision) has been selected (or a smaller eye was sufficient and selection has acted to decrease the eye scaling slope). Steeper slopes for wing size and antenna size would imply relative increases in the importance of flight ability and olfactory sensitivity [20,21]. Though our data are not sufficient to indicate that selection has acted to change the organ scaling slopes among these bumblebee colonies, they do indicate that it is possible to generate the variation needed for selection to act, if colonies are experiencing different environmental conditions.

The allometric variability found in this and other studies [3,11,12,14] may contribute to intraspecific differences in colony fitness because morphology affects worker foraging capability [4–7,9,21,24], which may change with body size as scaling relationships change.

There is already good evidence that foraging behaviour differs among bumblebee colonies [15,17] and what causes these differences could be explained through variability in allometric relationships among nests. Smith *et al.* 2016 [17] show that there are differences in the proportion of nectar forgers among *B. terrestris* colonies. There are also differences in the proportion of flower specialists and generalists [17], and none of these differences are explained through differences in median body size. It is also known that olfactory sensitivity is determined by antennal length [7], the scaling of which is different between colonies (figures 2 and 3). Therefore, differences in foraging behaviour might be driven by differences in allometric scaling.

The extent to which the observed differences in colony scaling might impact colony fitness can only be determined by comparative behavioural assays that establish differences in foraging efficiency that correlate with differences in colony scaling rules. The results of such a study would also provide a clue as to whether allometric differences among nests are adaptive or the result of developmental noise [46]. Furthermore, if the allometric differences presented here exist among colonies in naturally occurring populations [14], it could mediate temporal or spatial resource partitioning.

Resource partitioning based on sensory capability is well established among social insects, both intra- and interspecifically. In Australian *Myrmecia* ants, different castes and different species are active at different times of the day depending on their visual capabilities [47]. Stingless bees have a particularly steep allometric slope for their eyes, because this provides their workers with as large an eye as possible, enabling the largest workers to forage earlier in the day before nectar resources are depleted [5]. Larger bumblebees forage earlier than smaller conspecifics [8], likely due to larger bees having larger eyes [29]. Therefore, the differences we document in eye scaling slope among bumblebee colonies could be a driver of differential activity, due to some colonies having relatively larger eyes than others.

Differences in organ scaling among colonies in the same population may also contribute to resources in different spaces being exploited. Different species of *Drosophila* use different microhabitats depending on their relative sensory investment [48]. A similar phenomenon could cause differential microhabitat use by sympatric colonies if they experience differential slope changes (e.g. colonies four versus eight and four versus three, antenna scaling). Therefore, differences in the rules that determine worker organ scaling may contribute to differences in worker behaviour and resource partitioning, which will subsequently affect colony fitness [3].

## 4.3. Sensory and non-sensory allometry

When averaged across all colonies, we also found a common slope for sensory organs (eyes and antennae) that differed from the slope of the locomotory organ (wings) (figure 4). Though this relationship is not the same when examining scaling on a colony-by-colony basis, it does raise the question of whether sensory organs are following a different set of scaling rules from non-sensory organs. Differential expression of insulin-like receptors (ILR) on imaginal discs is a mechanism by which the steepness of a scaling slope among different organs can be varied [49]. Having a fixed expression of ILR for sensory imaginal discs and another level of expression of non-sensory structures would explain the observed scaling relationships. Further investigation is needed to establish if this pattern extends to other organs.

# 5. Conclusion

Intraspecific flexibility in allometric coefficients is prevalent in insects, with documented changes in scaling relationships reported due to nutrition [50,51] and temperature [50], as well as differences among and within populations [3,14,52]. Our data support these conclusions, which must naturally be the case if allometric relationships within a species are to be subject to natural selection [53,54]. The question remains as to whether this variability is generated randomly, is directed by genetic differences, is in response to environmental fluctuations and is present in naturally occurring populations [14].

Our results, in conjunction with others, show that morphological scaling among social insect colonies is not a fixed set of rigid developmental rules that are strictly adhered to within a given species. Among *B. terrestris* colonies, not only do we observe variability in the scaling of multiple organs, but we also observe putative trade-offs in organ investment. These allometric differences, whether adaptive or not, provide an alternative cause for observed differences in colony behaviour. Thus, our study provides impetus to examine new hypotheses concerning differential colony fitness and the constraints on social insect behaviour.

Data accessibility. Raw data and R code can be found at https://zenodo.org/record/4013716.

The data are provided in the electronic supplementary material [55].

Authors' contributions. C.D.P.: conceptualization, data curation, formal analysis, investigation, methodology, validation, visualization, writing—original draft and writing—review and editing; Z.B.J.: conceptualization, data curation, investigation, methodology and writing—review and editing; V.W.J.: investigation and writing—review and editing; Z.M.: investigation and writing—review and editing; M.G.: investigation and writing—review and editing; C.E.R.: investigation and writing—review and editing; A.M.: investigation and writing—review and editing; E.B.: conceptualization, funding acquisition, investigation, methodology, project administration, resources, supervision, validation, writing—review and editing.

All authors gave final approval for publication and agreed to be held accountable for the work performed therein.

Competing interests. We declare we have no competing interests.

Funding. Research was supported by grants to E.B. from the Swiss Research Council (Beamtime nos.: 20191425, 20190641), Vetenskaprådet (RGP0002/2017) and Human Frontiers Science Program (2018-06238).

Acknowledgements. Many thanks to Julia Meneghello and Fatih Aksoy for their hard work measuring compound eyes with AMIRA and to Tunhe Zhou and Jenny Romell for their invaluable micro-CT expertise. Thanks also to Signe Hägglund, Alexandra Hagelin, Vera Ranow, Viktoria Köppä for help measuring specimens.

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
