## [Peer Review File · Royal Society Open Science]

Review History

RSOS-201787.R0 (Original submission)

Review form: Reviewer 1

Is the manuscript scientifically sound in its present form?

Yes

Are the interpretations and conclusions justified by the results?

No

Is the language acceptable?

Yes

Do you have any ethical concerns with this paper?

No

Have you any concerns about statistical analyses in this paper?

No

Recommendation?

Major revision is needed (please make suggestions in comments)

Comments to the Author(s)

Please see the attached file for comments (Appendix A).

Review form: Reviewer 2**Is the manuscript scientifically sound in its present form?**

No

Are the interpretations and conclusions justified by the results?

No

Is the language acceptable?

Yes

Do you have any ethical concerns with this paper?

No

Have you any concerns about statistical analyses in this paper?

Yes

Recommendation?

Major revision is needed (please make suggestions in comments)

Comments to the Author(s)

Royal society review

This paper addressed an area of allometric research that is growing in importance - that of intraspecific differences in allometric scaling relationships.

Although well-written, I have a number of concerns, specifically regarding the reporting of the statistical analyses and the structure of the discussion.

In particular, I could not follow how the data analysis was undertaken, having looked at the method and the accompanying R code. Both of these should be improved. Model formula should be described and justified in the methods.

i.e. We specified a model of $y \sim x_1 \dots$ in interaction with x_2 and so on. Also, I am unconvinced of significant differences between hives having reviewed the R code. I think this is where the issues with the discussion arise.

There are few differences between colonies or only between specific colonies and the experiment has no way of knowing where these differences come from except chance. This results in the discussion being highly speculative and overly long.

I think the fact that allometric scaling differences exist between colonies is important to acknowledge but ultimately this paper should be revised into a shorter format to reflect what it can tell us, rather than discussing in great detail what it can't.

Abstract:

L15-16: Perhaps better to state that intraspecific allometric differences are understudied relative to broader (interspecific differences)

L21-L22: do you test for differences between colony sources? and you only account for differing temperatures with a random effect. I would remove the second half of this sentence and just state: we document significant allometric variability: two colonies had wing vs. antenna slope shifts...

Introduction

L43; Add reference

L85: I think this example needs more detail - i.e. shifts in tongue length intercepts (colonies had different tongue lengths per unit body mass) but the same slope

L109 - 116: This paragraph is not necessary in the introduction - it is better suited for the abstract or discussion. therefore I would remove it.

Methods

Statistics

I think this entire section needs a comprehensive edit and restructuring.

The models you present in your results section are not adequately described in the methods section. the model fixed and random effects (as described in table 1) should be presented in the method section.

you should also provide rationale for the selection of your random effects. Also, It is very uncertain how you arrived at your 'minimum adequate model'. And the accompanying R code does not demonstrate any model comparisons except for the full model and the 'minimum model'.

A cursory look at your dataset suggests this is not the most parsimonious model - indeed, quick but uninformed model selection suggests a model without hive "log10(length) ~ body_part + log_mass + (1 | temp/file_name) +

body_part:log_mass" is better (based off AIC). However, I understand you are interested in differences between hives so I would revise and think about how best to arrive at the 'best model'.

Is model selection really necessary? Perhaps you could just state your hypothesis - differences between colonies - and describe this.

Also, I believe the factor analysis should either be described in more detail or removed as I am uncertain how much it adds in terms of explaining allometric differences between colonies than the mixed effect models. Are there biological reasons to expect the observed clustering? Or is this just random?

And what does this tell us about intraspecific allometric scaling? that there is convergence in deviations in intercepts/slopes of allometric relationships. Are the clusters captured by the contrasts derived from mixed models? Also, if part of the aim was to compare allometric slopes between colonies, this can be done using the emtrends function in emmeans.

- perhaps this would be a simpler approach to address the same question.

Discussion

I find the discussion is too long (8 pages) and overly speculative and i recommend it is considerably edited prior to publication.

I do not think speculation is wrong, rather it is the result of allometric differences between observed between colonies when we do not know the reason for them occurring. However, I think the authors should consider which points are most important and discuss these. For example, the two paragraphs about temperature and genetic relatedness among commercial colonies are unnecessary in my opinion.

Decision letter (RSOS-201787.R0)

Dear Dr Perl

The Editors assigned to your paper RSOS-201787 "SUBSTANTIAL VARIABILITY IN MORPHOLOGICAL SCALING AMONG BUMBLEBEE COLONIES" have made a decision based on their reading of the paper and any comments received from reviewers.

Regrettably, in view of the reports received, the manuscript has been rejected in its current form. However, a new manuscript may be submitted which takes into consideration these comments.

We invite you to respond to the comments supplied below and prepare a resubmission of your manuscript. Below the referees' and Editors' comments (where applicable) we provide additional requirements. We provide guidance below to help you prepare your revision.

Please note that resubmitting your manuscript does not guarantee eventual acceptance, and we do not generally allow multiple rounds of revision and resubmission, so we urge you to make every effort to fully address all of the comments at this stage. If deemed necessary by the Editors, your manuscript will be sent back to one or more of the original reviewers for assessment. If the original reviewers are not available, we may invite new reviewers.

Please resubmit your revised manuscript and required files (see below) no later than 23-Sep-2021. Note: the ScholarOne system will 'lock' if resubmission is attempted on or after this deadline. If you do not think you will be able to meet this deadline, please contact the editorial office immediately.

Please note article processing charges apply to papers accepted for publication in Royal Society Open Science (<https://royalsocietypublishing.org/rsos/charges>). Charges will also apply to papers transferred to the journal from other Royal Society Publishing journals, as well as papers submitted as part of our collaboration with the Royal Society of Chemistry (<https://royalsocietypublishing.org/rsos/chemistry>). Fee waivers are available but must be requested when you submit your manuscript (<https://royalsocietypublishing.org/rsos/waivers>).

Thank you for submitting your manuscript to Royal Society Open Science and we look forward to receiving your resubmission. If you have any questions at all, please do not hesitate to get in touch.

Best regards,
Lianne Parkhouse
Editorial Coordinator

on behalf of Dr Sean Rands (Associate Editor) and Kevin Padian (Subject Editor)
openscience@royalsociety.org

Editor Comments to Author:

Thanks for your submission, which is a very careful and well-conducted study. As you can see, the reviewers appreciate its strengths and also have some (different) concerns, which they have brought out in well-articulated comments. One reviewer is concerned about inferences made on the basis of commercial species, and suggests modifying language about evolutionary implications. (Perhaps there are studies that show no significant differences between wild and commercial populations in such regards?) Concerns about the vocabulary of "scaling" and "allometry" are well put, although traditionally some authors have considered isometry simply an absence of allometry (as Hardy-Weinberg equilibrium is a lack of selection), so I ask that you consider the question in that light.

The other reviewer is concerned about the statistical analysis, and here it may be that you will need some time to reassess. For this reason primarily I am logging a "reject/resub" decision because a "major revision" decision is more constraining in time. Please attend to the comments individually and we look forward to your resubmission taking into account the concerns of the reviewers.

Also, may I ask that you change the manuscript font to something more standard such as Times New Roman or Palatino? It is a bit difficult to read and hard to distinguish italicized words from normal font. Thanks very much.

Reviewer comments to Author:

Reviewer: 1
Comments to the Author(s)
Please see the attached file for comments

Reviewer: 2
Comments to the Author(s)
Royal society review

This paper addressed an area of allometric research that is growing in importance - that of intraspecific differences in allometric scaling relationships. Although well-written, I have a number of concerns, specifically regarding the reporting of the statistical analyses and the structure of the discussion.

In particular, I could not follow how the data analysis was undertaken, having looked at the method and the accompanying R code. Both of these should be improved. Model formula should be described and justified in the methods.
i.e. We specified a model of $y \sim x_1 \dots$ in interaction with x_2 and so on. Also, I am unconvinced of significant differences between hives having reviewed the R code. I think this is where the issues with the discussion arise.

There are few differences between colonies or only between specific colonies and the experiment has no way of knowing where these differences come from except chance. This results in the discussion being highly speculative and overly long.

I think the fact that allometric scaling differences exist between colonies is important to acknowledge but ultimately this paper should be revised into a shorter format to reflect what it can tell us, rather than discussing in great detail what it can't.

Abstract:

L15-16: Perhaps better to state that intraspecific allometric differences are understudied relative to broader (interspecific differences)

L21-L22: do you test for differences between colony sources? and you only account for differing temperatures with a random effect. I would remove the second half of this sentence and just state: we document significant allometric variability: two colonies had wing vs. antenna slope shifts...

Introduction

L43; Add reference

L85: I think this example needs more detail - i.e. shifts in tongue length intercepts (colonies had different tongue lengths per unit body mass) but the same slope

L109 - 116: This paragraph is not necessary in the introduction - it is better suited for the abstract or discussion. therefore I would remove it.

Methods

Statistics

I think this entire section needs a comprehensive edit and restructuring.

The models you present in your results section are not adequately described in the methods section. the model fixed and random effects (as described in table 1) should be presented in the method section.

you should also provide rationale for the selection of your random effects. Also, It is very uncertain how you arrived at your 'minimum adequate model'. And the accompanying R code does not demonstrate any model comparisons except for the full model and the 'minimum model'.

A cursory look at your dataset suggests this is not the most parsimonious model - indeed, quick but uninformed model selection suggests a model without hive "log10(length) ~ body_part + log_mass + (1 | temp/file_name) +

body_part:log_mass" is better (based off AIC). However, I understand you are interested in differences between hives so I would revise and think about how best to arrive at the 'best model'.

Is model selection really necessary? Perhaps you could just state your hypothesis - differences between colonies - and describe this.

Also, I believe the factor analysis should either be described in more detail or removed as I am uncertain how much it adds in terms of explaining allometric differences between colonies than the mixed effect models. Are there biological reasons to expect the observed clustering? Or is this just random?

And what does this tell us about intraspecific allometric scaling? that there is convergence in deviations in intercepts/slopes of allometric relationships. Are the clusters captured by the contrasts derived from mixed models? Also, if part of the aim was to compare allometric slopes between colonies, this can be done using the emtrends function in emmeans.

- perhaps this would be a simpler approach to address the same question.

Discussion

I find the discussion is too long (8 pages) and overly speculative and i recommend it is considerably edited prior to publication.

I do not think speculation is wrong, rather it is the result of allometric differences between observed between colonies when we do not know the reason for them occurring. However, i think the authors should consider which points are most important and discuss these. For example, the two paragraphs about temperature and genetic relatedness among commercial colonies are unnecessary in my opinion.

===PREPARING YOUR MANUSCRIPT===

===PREPARING YOUR REVISION IN SCHOLARONE===

Author's Response to Decision Letter for (RSOS-201787.R0)

See Appendix B.

RSOS-211436.R0

Review form: Reviewer 1

Is the manuscript scientifically sound in its present form?

Yes

Are the interpretations and conclusions justified by the results?

Yes

Is the language acceptable?

Yes

Do you have any ethical concerns with this paper?

No

Have you any concerns about statistical analyses in this paper?

No

Recommendation?

Accept with minor revision (please list in comments)

Comments to the Author(s)

See attached (Appendix C).

Decision letter (RSOS-211436.R0)

Dear Dr Perl

On behalf of the Editors, we are pleased to inform you that your Manuscript RSOS-211436 "SUBSTANTIAL VARIABILITY IN MORPHOLOGICAL SCALING AMONG BUMBLEBEE COLONIES" has been accepted for publication in Royal Society Open Science subject to minor revision in accordance with the referees' reports. Please find the referees' comments along with any feedback from the Editors below my signature.

We invite you to respond to the comments and revise your manuscript. Below the referees' and Editors' comments (where applicable) we provide additional requirements. Final acceptance of

your manuscript is dependent on these requirements being met. We provide guidance below to help you prepare your revision.

Please submit your revised manuscript and required files (see below) no later than 7 days from today's (ie 10-Dec-2021) date. Note: the ScholarOne system will 'lock' if submission of the revision is attempted 7 or more days after the deadline. If you do not think you will be able to meet this deadline please contact the editorial office immediately.

on behalf of Dr Sean Rands (Associate Editor) and Kevin Padian (Subject Editor)
openscience@royalsociety.org

Editor comments:

Thanks for your resubmission. Our reviewer was satisfied and had only a few minor suggestions, which I hope you'll incorporate. Best wishes.

Reviewer comments to Author:

Reviewer: 1

Comments to the Author(s)

See attached

===PREPARING YOUR MANUSCRIPT===

one version should clearly identify all the changes that have been made (for instance, in coloured highlight, in bold text, or tracked changes);

Please ensure that you include an acknowledgements' section before your reference list/bibliography. This should acknowledge anyone who assisted with your work, but does not

qualify as an author per the guidelines at <https://royalsociety.org/journals/ethics-policies/openness/>.

===PREPARING YOUR REVISION IN SCHOLARONE===

- Ensure that your data access statement meets the requirements at <https://royalsociety.org/journals/authors/author-guidelines/#data>. You should ensure that you cite the dataset in your reference list. If you have deposited data etc in the Dryad repository, please only include the 'For publication' link at this stage. You should remove the 'For review' link.
- If you are requesting an article processing charge waiver, you must select the relevant waiver option (if requesting a discretionary waiver, the form should have been uploaded, see 'File upload' above).
- If you have uploaded any electronic supplementary (ESM) files, please ensure you follow the guidance at <https://royalsociety.org/journals/authors/author-guidelines/#supplementary-material> to include a suitable title and informative caption. An example of appropriate titling and captioning may be found at https://figshare.com/articles/Table_S2_from_Is_there_a_trade-off_between_peak_performance_and_performance_breadth_across_temperatures_for_aerobic_scope_in_teleost_fishes_/3843624.

Author's Response to Decision Letter for (RSOS-211436.R0)

See Appendix D.

Decision letter (RSOS-211436.R1)

Dear Dr Perl,

I am pleased to inform you that your manuscript entitled "SUBSTANTIAL VARIABILITY IN MORPHOLOGICAL SCALING AMONG BUMBLEBEE COLONIES" is now accepted for publication in Royal Society Open Science.

Please note that "erensto.restrepo@zoologi.su.se" is not accepting messages from the journal.
Please can you confirm a correct email address or an alternative active contact address?

on behalf of Dr Sean Rands (Associate Editor) and Kevin Padian (Subject Editor)
openscience@royalsociety.org

Appendix A

The authors investigated organ to body size scaling relationships across bumblebee colonies from commercial sources. They found significant variation in organ scaling between bumblebee colonies and suggest that this may allow for differences in behavior, particularly in spatiotemporal foraging patterns. Although I suggest that the authors approach any evolutionary or adaptive explanations or conclusions based on commercially-bred bumblebees with great caution, I do think that this study poses a significant contribution in the field. It is rare for a study to consider intraspecific differences in morphological scaling, and I believe that this, along with a discussion of its implications, has value. Nonetheless, there are several areas where substantial improvements can be made, detailed below.

General comments:

1) I am concerned with the evolutionary and natural selection-based conclusions that the authors make based on data collected from commercially-bred bumblebees. I agree that the *potential* for variation among and within bumblebee colonies is evident, based on the data presented. However, because the colonies are commercially reared, this study does not provide any evidence of similar levels of natural variation in the field. Thus, I believe that the claims that natural selection is or has acted on the morphology of these individuals and colonies to generate the observed variation is unfounded. Some supporting data or evidence from field colonies that are more directly subject to natural selection would greatly complement this study.

2) I find some of the key terms and phrases that the authors use frequently throughout the manuscript to be ill-defined or problematic:

- “developmental instability, noise, and/or optimum”: This terminology needs to be clearly defined.
- “allometry” or “allometric” when referring to the field or methodology: I understand that this terminology has often been used in this way. However, the terms “allometric” and “isometric” have mutually exclusive definitions. Despite this, the authors refer to the field of “allometry,” which encompasses both terms/morphological variants. I find distinguishing between the two definitions of “allometry” throughout the manuscript to be challenging. I would suggest replacing “allometry” or “allometric” when referring to the field of study or methodology with “scaling,” or something similar. For example, instead of starting the abstract with the phrase “organ allometry,” which could be incorrectly assumed to mean “disproportionate scaling of organs” by a reader, I would suggest the phrase “organ scaling.”
- “rules”: This terminology is fine when referring to the idea that morphological scaling is subject to certain rules, an idea that this manuscript provides evidence against. However, since this study reveals much individual- and colony-level variation, this terminology is more out of place when referring to the results of the study and the different “rules” that colonies are following. For example, the usage of this terminology is appropriate in lines 410-412, but less appropriate in lines 412-414.

- scaling relationships at “higher levels of organisation” or “broader morphological investigations”: It is unclear what is meant by these phrases or what scale the authors are referring to.

- “slope shifts”: This terminology needs to be clearly defined or should be replaced with something more intuitive – I think the authors simply mean “differences in slope.”

3) I found the writing to be quite vague in some parts of the manuscript. I have identified the most problematic sections in the line-specific comments below.

Abstract:

L18-20: This reads as if you investigated the relationships between each of the three organs, when instead you investigated the morphological scaling of each of the three organs *in relation to body size* and subsequently compared them. Scaling relationships between each of the organs might be an interesting addition to the manuscript, however.

L20-22: “despite the absence of environmental variability and their differing commercial sources”: In a few places in the manuscript, you mention that the colonies were reared at different temperatures. I would not consider this an absence of environmental variability, although I understand your claim that the temperature differences did not account for the morphological variability. Additionally, the idea of finding significant morphological variability *despite* differing commercial sources is confusing – wouldn’t variation based on commercial source be expected?

Introduction:

Overall: For a journal covering a broad range of science topics, this introduction would benefit from more background on social insect biology, particularly regarding colony vs. individual level fitness benefits/costs. For example, more background is needed in the paragraphs from lines 76-100. Background on the importance of worker allometry to task allocation or general colony functioning would also be useful. Many readers may not understand the role that worker variability plays in colony fitness.

L32: “The control of organ size is a central question in animal biology”: You do not appear to address this central question, which I would expect to require investigations into the physiological or developmental control of organ size. Thus, I would suggest removing this part of the sentence and simply begin the sentence with “The relative size of organs...” that you include later in the sentence.

L34-37: I suggest adding “, of higher abundance or nutritional quality.” after “superior resources.” A reference is needed for this idea. You can then split this sentence into two, starting the next sentence with “Therefore, organs in...”

L37-40: I suggest rewording this sentence and perhaps splitting it up for clarity. Something like: “The relationship between organ size and body size can be isometric (organ size scales proportionately with body size) or allometric (organ size scales disproportionately with body size). Organ allometry can be positive, in which organs are relatively larger per unit body size, or negative, in which organs are relatively smaller per unit body size.”

L40-42: I suggest rewording to “Changes in slope and intercept...” and moving this sentence from its current location to after the sentence ending with “differences in slope” on line 47. Also, this sentence is missing a reference, as the authors appear to indicate.

L53: replace “how it presents” with “these relationships”

L54-64: I find this portion of the paragraph very vague. Providing more explanations of these concepts or some concrete examples would be useful.

L65: You refer specifically to morphological variability “within a species” here, but the concepts that you discuss later in the paragraph can be just as easily applied to variability across species and many other levels of biological organization. If there are concepts unique to the intraspecific level, elaborate on them here. Otherwise, I suggest cutting “within a species.”

L67-69: A reference is needed to support this idea.

L73-75: Is this statement referring to only bumblebees, all social insects, or insects in general? If the former, provide a reference to support this claim. If the latter, a reference and/or concrete example would be useful.

L80: replace “body” with “organ”

L81: start a new sentence at “therefore,...”

L84: insert “intraspecific” after “explored” and “social insect” after “among”

L97-98: rephrase to: “... exist in foraging, learning, the degree of worker specialisation, and aggression.”

L98-100: I assume you mean that this phenomenon is unknown in bumblebees. If so, make this clear.

L109-110: replace “how” with “organ scaling among”; remove “scale their organs”

L109-116: Due to the presentation of results and the interpretive nature of this paragraph, I think it would be a great opening paragraph for the discussion!

Methods:

L123-124: I suggest adding information about which colonies were reared at each of the two temperatures, at least in the supplemental materials. Was this based on the commercial source of the colonies?

L124: replace “climate control” with “climate-controlled”

L130: remove “individual”

L135-136: Perhaps describe what equipment you used to photograph the antennae, as you do below for the forewings.

L146: remove “subsequent”

L158-159: An explanation for why you used different magnifications, depending on the colony, is needed.

L161-164: References are needed for these methods.

Results:

- Only 5 out of 12 colonies are included in the within- and across-colony scaling relationship results. More information is needed for the reader to gain a broader understanding of your results: 1) Within colonies, are the colonies or scaling relationships that you don't explicitly mention all isometric or allometric? Are the slopes and intercepts the same for each organ for these unmentioned colonies? 2) Can the reader assume that any across colony comparisons that are not explicitly mentioned have similar/identical scaling relationships? 3) What did the scaling relationships look like depending on temperature treatments or commercial sources? 4) Was mean body size different across colonies, temperature treatments, or commercial sources? This is particularly relevant to the information reported in L225-229. 5) It would be good to report R^2 values for your regression lines to get an idea of the degree of within colony variation. This could be reported as a table in your supplemental materials, with notable variation mentioned in the main text of the results.
- I suggest rearranging the order in which you present your mixed effects models, starting with broader relationships first. For example, it's easier to orient the reader if you present the colony-level scaling data before you then compare this data across colonies. Here is a suggested order of sections, based on your subheadings: 1) organ size independent of organ identity, 2) mean organ size (intercept) across colonies, 3) organ slopes independent of colony affiliation, 4) comparing slopes of organs within colonies, 5) comparing slopes of different organs across colonies.

L211-224: The written explanation of the scaling relationships in this section is less concise than in the previous section. For example, “for a unit increase in body size of bees from colony three, their wings will get proportionally larger than their antennae” can be rewritten as “as body size increases, wing size increases faster than antennae size,” which is more concise and similar to the previous section's writing.

L240-241: replace “negative allometry. Smaller” with “negative allometry, meaning smaller”

Discussion:

L258: The phrase “There exist mechanisms for generating allometric flexibility within a species” is very vague and requires more explanation.

L259: replace “will” with “can”

L262-264: The sentence starting with “We compared...” is redundant with the previous sentence. I suggest you cut it.

L270: The reader will likely not remember the specific differences observed in colony nine. Restate these differences here.

L286-294: This information appears to be restating the results in a similar level of detail. I suggest summarizing and interpreting this information here instead.

L301-307: I was not aware that bumblebees possessed different morphological worker castes, and I don't know of any published data to support this idea. Are you claiming that your data provides evidence for the presence of morphological worker castes in bumblebees? If so, what is the basis of the definition of worker castes you are using here? I would find such a conclusion based on the data presented to be very problematic. Alternatively, do you simply mean that these colonies have different proportions of *differently-sized* workers?

L308-318: Please refer to the first general comment above. I find making conclusions about the adaptiveness of the observed variability in scaling relationships based on commercially-bred bumblebee colonies to be very problematic.

L331-332: replace “which will change if the relationship between organ and body size changes” with “which may change with body size as scaling relationships change”

L350-361: Here, the authors use evidence from two across-species studies to draw conclusions about potential differences in across-colony intraspecific behavior. This should be approached with great caution. There may be other differences across species, aside from variation in organ scaling, that account for spatiotemporal differences in foraging. Additionally, the scale of this variation may be very different in the cited studies than in the current study.

L392-393: The idea that feeding the colonies *ad libitum* removed any variation in nutrition that could account for differences in morphology is flawed, particularly in light of the authors' previous claims that differences in morphology will lead to differences in foraging behavior and resource exploitation.

Table 1: I think this table is well suited to the supplemental material.

Table 3: I suggest reducing the number of significant figures to 2, to be consistent with Table 2.

Figure 1: I think this figure is well suited to the supplemental material.

Figure 2: The colors in this figure are very difficult to distinguish. Perhaps the authors could use a combination of colors, solid vs. dashed lines, and point shapes instead.

Figures 4 & 5: The use of some of the same colors for these figures as those used in Figure 2, in which the colors represent colony identity, is confusing. I suggest using entirely different colors. Alternatively, Figure 4 does not require different colors to be clear, and the authors could use different point shapes to represent clusters in Figure 5.

Appendix B

Manuscript ID RSOS-201787 entitled "Substantial variability in morphological scaling among bumblebee colonies".

Editor Comments to Author(s):

Thanks for your submission, which is a very careful and well-conducted study. As you can see, the reviewers appreciate its strengths and also have some (different) concerns, which they have brought out in well-articulated comments. One reviewer is concerned about inferences made on the basis of commercial species, and suggests modifying language about evolutionary implications. (Perhaps there are studies that show no significant differences between wild and commercial populations in such regards?) Concerns about the vocabulary of "scaling" and "allometry" are well put, although traditionally some authors have considered isometry simply an absence of allometry (as Hardy-Weinberg equilibrium is a lack of selection), so I ask that you consider the question in that light. The other reviewer is concerned about the statistical analysis, and here it may be that you will need some time to reassess. For this reason primarily I am logging a "reject/resub" decision because a "major revision" decision is more constraining in time. Please attend to the comments individually and we look forward to your resubmission taking into account the concerns of the reviewers. Also, may I ask that you change the manuscript font to something more standard such as Times New Roman or Palatino? It is a bit difficult to read and hard to distinguish italicized words from normal font. Thanks very much.

We would like to thank the editor and the reviewers for their thoughtful and constructive comments and suggestions. We have addressed the individual comments below.

Comments to Author(s):

Referee 1:

The authors investigated organ to body size scaling relationships across bumblebee colonies from commercial sources. They found significant variation in organ scaling between bumblebee colonies and suggest that this may allow for differences in behavior, particularly in spatiotemporal foraging patterns. Although I suggest that the authors approach any evolutionary or adaptive explanations or conclusions based on commercially-bred bumblebees with great caution, I do think that this study poses a significant contribution in the field. It is rare for a study to consider intraspecific differences in morphological scaling, and I believe that this, along with a discussion of its implications, has value. Nonetheless, there are several areas where substantial improvements can be made, detailed below.

General comments:

1. I am concerned with the evolutionary and natural selection-based conclusions that the authors make based on data collected from commercially-bred bumblebees. I agree that the potential for variation among and within bumblebee colonies is evident, based on the data presented. However, because the colonies are commercially reared, this study does not provide any evidence of similar levels of natural variation in the field. Thus, I believe that the claims that natural selection is or has acted on the morphology of these individuals and colonies to generate the observed variation is unfounded. Some supporting data or evidence from field colonies that are more directly subject to natural selection would greatly

complement this study.

Though our data are not from naturally occurring populations, there are data from natural bumblebee (doi:10.1046/j.1420-9101.1995.8060725.x) and ant (doi:10.1038/srep24204) populations that have documented allometric differences among colonies. Both these articles are cited in the manuscript. We have also modified the conclusion to highlight the utility of searching for these relationships among non-commercial colonies.

2. I find some of the key terms and phrases that the authors use frequently throughout the manuscript to be ill-defined or problematic:

- “developmental instability, noise, and/or optimum”: This terminology needs to be clearly defined.

These terms have either been revised or removed from the manuscript. A reference has been inserted to guide readers towards a suitable paper regarding developmental noise.

- “allometry” or “allometric” when referring to the field or methodology: I understand that this terminology has often been used in this way. However, the terms “allometric” and “isometric” have mutually exclusive definitions. Despite this, the authors refer to the field of “allometry,” which encompasses both terms/morphological variants. I find distinguishing between the two definitions of “allometry” throughout the manuscript to be challenging. I would suggest replacing “allometry” or “allometric” when referring to the field of study or methodology with “scaling,” or something similar. For example, instead of starting the abstract with the phrase “organ allometry,” which could be incorrectly assumed to mean “disproportionate scaling of organs” by a reader, I would suggest the phrase “organ scaling.” The suggested changes have been implemented.

- “rules”: This terminology is fine when referring to the idea that morphological scaling is subject to certain rules, an idea that this manuscript provides evidence against. However, since this study reveals much individual- and colony-level variation, this terminology is more out of place when referring to the results of the study and the different “rules” that colonies are following. For example, the usage of this terminology is appropriate in lines 410-412, but less appropriate in lines 412-414.

This has been modified as suggested

- scaling relationships at “higher levels of organisation” or “broader morphological investigations”: It is unclear what is meant by these phrases or what scale the authors are referring to.

These phrases have been respectively replaced with “across the entire species or population” and “Intraspecific allometric differences between social insect colonies are understudied relative to investigations into the effects of absolute organ size. However, intraspecific scaling could provide a source of behavioural variation that has yet to be recognised.”

- “slope shifts”: This terminology needs to be clearly defined or should be replaced with something more intuitive – I think the authors simply mean “differences in slope.”

The text has been modified as suggested

I found the writing to be quite vague in some parts of the manuscript. I have identified the most problematic sections in the line-specific comments below:

1. (Abstract, L18-20) This reads as if you investigated the relationships between each of the three organs, when instead you investigated the morphological scaling of each of the three

organs in relation to body size and subsequently compared them. Scaling relationships between each of the organs might be an interesting addition to the manuscript, however. This has now been clarified.

2. (Abstract, L20-22, “despite the absence of environmental variability and their differing commercial sources”) In a few places in the manuscript, you mention that the colonies were reared at different temperatures. I would not consider this an absence of environmental variability, although I understand your claim that the temperature differences did not account for the morphological variability. Additionally, the idea of finding significant morphological variability despite differing commercial sources is confusing – wouldn’t variation based on commercial source be expected?

We thank the reviewer for this comment. We have amended the abstract to make it clear that although there was environmental variability, it was not sufficient to explain the variability we observed in the scaling relationships. We would expect colonies of different sources to be more similar to each other in their scaling relationships, however, this was not born out by the factorial analysis of mixed data. We have also been advised to remove much of the speculation regarding the commercial sources of the colonies.

3. (Introduction, Overall) For a journal covering a broad range of science topics, this introduction would benefit from more background on social insect biology, particularly regarding colony vs. individual level fitness benefits/costs. For example, more background is needed in the paragraphs from lines 76- 100. Background on the importance of worker allometry to task allocation or general colony functioning would also be useful. Many readers may not understand the role that worker variability plays in colony fitness.

We have added some additional detail to the introduction as well as restructuring it, so that this information is clearer.

4. (Introduction, L32, “The control of organ size is a central question in animal biology”) You do not appear to address this central question, which I would expect to require investigations into the physiological or developmental control of organ size. Thus, I would suggest removing this part of the sentence and simply begin the sentence with “The relative size of organs...” that you include later in the sentence.

This has been revised as suggested.

5. (Introduction, L34-37) I suggest adding “, of higher abundance or nutritional quality.” after “superior resources.” A reference is needed for this idea. You can then split this sentence into two, starting the next sentence with “Therefore, organs in...”

This has been revised as suggested.

6. (Introduction, L37-40) I suggest rewording this sentence and perhaps splitting it up for clarity. Something like: “The relationship between organ size and body size can be isometric (organ size scales proportionately with body size) or allometric (organ size scales disproportionately with body size). Organ allometry can be positive, in which organs are relatively larger per unit body size, or negative, in which organs are relatively smaller per unit body size.”

This has been revised as suggested.

7. (Introduction, L40-42) I suggest rewording to “Changes in slope and intercept...” and moving this sentence from its current location to after the sentence ending with “differences in slope” on line 47. Also, this sentence is missing a reference, as the authors appear to

indicate.

This has been revised as suggested, reference added.

8. (Introduction, L53) replace “how it presents” with “these relationships”

This has been revised as suggested.

9. (Introduction, L54-64) I find this portion of the paragraph very vague. Providing more explanations of these concepts or some concrete examples would be useful.

The section has been removed and modified

10.(Introduction, L65) You refer specifically to morphological variability “within a species” here, but the concepts that you discuss later in the paragraph can be just as easily applied to variability across species and many other levels of biological organization. If there are concepts unique to the intraspecific level, elaborate on them here. Otherwise, I suggest cutting “within a species.”

“Within a species” has been removed.

11. (Introduction, L67-69) A reference is needed to support this idea.

This sentence has been removed.

12. (Introduction, L73-75) Is this statement referring to only bumblebees, all social insects, or insects in general? If the former, provide a reference to support this claim. If the latter, a reference and/or concrete example would be useful.

*“Owen and Harder (1995) found significant shifts in tongue length intercept, but not slope, among different colonies of *Bombus huntii* and *B. occidentalis*. They concluded that these differences are heritable and may influence colony fitness.”*

This is the line from the original manuscript, at least the one in our possession. I don't suspect this is actually the sentence the referee is querying, as this is specifically regarding certain species of bumblebee and is already cited. If the sentence the referee refers to remains in the manuscript, please let us know and we will gladly modify as requested.

13. (Introduction, L80) Replace “body” with “organ”

Replaced.

14. (Introduction, L81) Start a new sentence at “therefore,...”

Revised as suggested.

15. (Introduction, L84) Insert “intraspecific” after “explored” and “social insect” after “among”

Inserted.

16. (Introduction, L97-98) Rephrase to: “... exist in foraging, learning, the degree of worker specialisation, and aggression.”

Rephrased as suggested.

17. (Introduction, L98-100) I assume you mean that this phenomenon is unknown in bumblebees. If so, make this clear.

Clarified.

18. (Introduction, L109-110) Replace “how” with “organ scaling among”; remove “scale

their organs”

Revised as suggested.

19. (Introduction, L109-116) Due to the presentation of results and the interpretive nature of this paragraph, I think it would be a great opening paragraph for the discussion!

Removed from the Introduction, and moved to the discussion.

20. (Methods, L123-124) I suggest adding information about which colonies were reared at each of the two temperatures, at least in the supplemental materials. Was this based on the commercial source of the colonies?

The split in temperature was not based on commercial sources, examples from each source were placed into both temperatures. The requested information has been added to the methods and to the supplement.

21. (Methods, L124) Replace “climate control” with “climate-controlled”

Revised as suggested.

22. (Methods, L130) Remove “individual”

Revised as suggested.

23. (Methods, L135-136) Perhaps describe what equipment you used to photograph the antennae, as you do below for the forewings.

Revised as suggested.

24. (Methods, L146) Remove “subsequent”

Revised as suggested.

25. (Methods, L158-159) An explanation for why you used different magnifications, depending on the colony, is needed.

An explanation has been added. It was due to differences in the setup between beamtimes, we were experimenting with the settings in order to find the optimal result with the scans.

26. (Methods, L161-164) References are needed for these methods.

Appropriate references have been added.

27. (Results) Only 5 out of 12 colonies are included in the within- and across-colony scaling relationship results. More information is needed for the reader to gain a broader understanding of your results:

a) Within colonies, are the colonies or scaling relationships that you don’t explicitly mention all isometric or allometric? Are the slopes and intercepts the same for each organ for these unmentioned colonies?

We apologise for omitting this information. Slopes and intercepts that are not explicitly mentioned in the results section were non-significantly different from each other.

b) Can the reader assume that any across colony comparisons that are not explicitly mentioned have similar/identical scaling relationships?

Likewise, colonies that were not mentioned were not significantly different from each other and were excluded for succinctness.

c) What did the scaling relationships look like depending on temperature treatments or

commercial sources?

This investigation, whilst acknowledging the conditions that the bees were reared in, was never intended to be about temperature effects and therefore we did not include any analysis on this. Similarly, we considered the provenance of the colonies as a source of allometric variability but did not explicitly include this factor in our models due to constraints of sample size. One of the major revisions we were given indicated that the focus on colony provenance was spurious and much of this speculation has been subsequently omitted.

d) Was mean body size different across colonies, temperature treatments, or commercial sources? This is particularly relevant to the information reported in L225-229. Please see the above comment in c) regarding our position on the temperature treatments and commercial sources. Two colonies, 2 and 11, had a higher mean fresh body mass than the other colonies. This information has been added to the results section.

e) It would be good to report R^2 values for your regression lines to get an idea of the degree of within colony variation. This could be reported as a table in your supplemental materials, with notable variation mentioned in the main text of the results.

We appreciate that R^2 values would be useful, however, we feel that Figure 3 already provides a good indication of the within nest variability.

28. (Results) I suggest rearranging the order in which you present your mixed effects models, starting with broader relationships first. For example, it's easier to orient the reader if you present the colony-level scaling data before you then compare this data across colonies. Here is a suggested order of sections, based on your subheadings: 1) organ size independent of organ identity, 2) mean organ size (intercept) across colonies, 3) organ slopes independent of colony affiliation, 4) comparing slopes of organs within colonies, 5) comparing slopes of different organs across colonies.

Revised as suggested.

29. (Results, L211-224) The written explanation of the scaling relationships in this section is less concise than in the previous section. For example, "for a unit increase in body size of bees from colony three, their wings will get proportionally larger than their antennae" can be rewritten as "as body size increases, wing size increases faster than antennae size," which is more concise and similar to the previous section's writing.

Revised, section has been made more succinct.

30. (Results, L240-241) Replace "negative allometry. Smaller" with "negative allometry, meaning smaller"

Revised as suggested.

31. (Discussion, L258) The phrase "There exist mechanisms for generating allometric flexibility within a species" is very vague and requires more explanation.

Sentence has been deleted to improve clarity and succinctness.

32. (Discussion, L259) Replace "will" with "can"

Revised as suggested.

33. (Discussion, L262-264) The sentence starting with "We compared..." is redundant with the previous sentence. I suggest you cut it.

Sentence removed.

34. (Discussion, L270) The reader will likely not remember the specific differences observed in colony nine. Restate these differences here.

Differences have been restated, added: “where wing size scaled with a steeper slope than both antenna length and eye area”.

35. (Discussion, L286-294) This information appears to be restating the results in a similar level of detail. I suggest summarizing and interpreting this information here instead.

Section removed.

36. (Discussion, L301-307) I was not aware that bumblebees possessed different morphological worker castes, and I don't know of any published data to support this idea. Are you claiming that your data provides evidence for the presence of morphological worker castes in bumblebees? If so, what is the basis of the definition of worker castes you are using here? I would find such a conclusion based on the data presented to be very problematic. Alternatively, do you simply mean that these colonies have different proportions of differently-sized workers?

This passage has been deleted to ensure the discussion is not overly long and to cut down on some of the speculation regarding the results.

37. (Discussion, L308-318) Please refer to the first general comment above. I find making conclusions about the adaptiveness of the observed variability in scaling relationships based on commercially-bred bumblebee colonies to be very problematic.

The portion of the discussion regarding adaptability and commercial origin has been removed.

38. (Discussion, L331-332) Replace “which will change if the relationship between organ and body size changes” with “which may change with body size as scaling relationships change”

Revised as suggested.

39. (Discussion, L350-361) Here, the authors use evidence from two across-species studies to draw conclusions about potential differences in across-colony intraspecific behavior. This should be approached with great caution. There may be other differences across species, aside from variation in organ scaling, that account for spatiotemporal differences in foraging. Additionally, the scale of this variation may be very different in the cited studies than in the current study.

We have added more detail and rewritten this section, it should now be clear that resource partitioning due to differences in sensory capability has been found within and across species, especially in social insects.

40. (Discussion, L392-393) The idea that feeding the colonies ad libitum removed any variation in nutrition that could account for differences in morphology is flawed, particularly in light of the authors' previous claims that differences in morphology will lead to differences in foraging behavior and resource exploitation.

We have amended this sentence to reflect the referee's insights:

“All colonies were provided with identical feeding regimes (fed ad libitum) suggesting that limitations on access to nutrition is unlikely to have been the cause of the observed allometric shifts.” All colonies were provided with more sugar water and pollen than needed, therefore

we think it is unlikely that nutrition was a limiting factor during development and can be excluded as a cause of the differences we observed in allometric scaling.

41. (Table 1) I think this table is well suited to the supplemental material.

Revised as suggested.

42. (Table 3) I suggest reducing the number of significant figures to 2, to be consistent with Table 2.

Revised as suggested.

43. (Figure 1) I think this figure is well suited to the supplemental material.

Revised as suggested.

44. (Figure 2) The colors in this figure are very difficult to distinguish. Perhaps the authors could use a combination of colors, solid vs. dashed lines, and point shapes instead.

We have amended the figure with colours and dashed lines. We recognise that it is difficult to distinguish lines and data on such a crowded plot, we hope that Fig. X indicates the differences in slope better.

45. (Figures 4 & 5) The use of some of the same colors for these figures as those used in Figure 2, in which the colors represent colony identity, is confusing. I suggest using entirely different colors. Alternatively, Figure 4 does not require different colors to be clear, and the authors could use different point shapes to represent clusters in Figure 5.

We have removed the colours from Figure 4 and added different shapes instead. The colours on Figure 5 have been changed to be different from those used in Figure 2.

Referee 2:

This paper addressed an area of allometric research that is growing in importance - that of intraspecific differences in allometric scaling relationships. Although well-written, I have a number of concerns, specifically regarding the reporting of the statistical analyses and the structure of the discussion.

General comments:

1. In particular, I could not follow how the data analysis was undertaken, having looked at the method and the accompanying R code. Both of these should be improved.

The method and the R code have been revised as suggested.

2. Model formula should be described and justified in the methods. i.e. We specified a model of $y \sim x_1 \dots$ in interaction with x_2 and so on.

This information has now been included in the main body of the text and in the supplement.

3. Also, I am unconvinced of significant differences between hives having reviewed the R code. I think this is where the issues with the discussion arise. There are few differences between colonies or only between specific colonies and the experiment has no way of knowing where these differences come from except chance.

We hope the changes made to the R code and the comments regarding the statistical methods below aid in clarifying the differences we observe in colony scaling relationships. Though we

do not have a mechanism to explain these differences, even if they are by chance, these differences still exist in opposition to a null hypothesis of all colonies scaling all their organs in the same way.

4. This results in the discussion being highly speculative and overly long. I think the fact that allometric scaling differences exist between colonies is important to acknowledge but ultimately this paper should be revised into a shorter format to reflect what it can tell us, rather than discussing in great detail what it can't.

We agree that the discussion was too long and contained too much speculation. The discussion has been significantly shortened and is more succinct.

Detailed comments:

1. (Abstract, L15-16) Perhaps better to state that intraspecific allometric differences are understudied relative to broader (interspecific differences)

Revised as suggested.

2. (Abstract, L21-L2) Do you test for differences between colony sources? and you only account for differing temperatures with a random effect. I would remove the second half of this sentence and just state: we document significant allometric variability: two colonies had wing vs. antenna slope shifts...

The abstract has been amended as follows: "Though colonies were exposed to different rearing temperatures, this environmental variability did not explain the variability of the scaling relationships."

3. (Introduction, L43) Add reference

The introduction has been amended, this sentence is no longer in the manuscript.

4. (Introduction, L85) I think this example needs more detail - i.e. shifts in tongue length intercepts (colonies had different tongue lengths per unit body mass) but the same slope. The sentence has been amended to reflect the suggestions of the referee: "Owen and Harder (1995) examined scaling among different colonies of *Bombus huntii* and *B. occidentalis*. They found significant shifts in intercept among colonies when regressing proboscis length against wing length but no differences in slope."

5. (Introduction, L109 – 116) This paragraph is not necessary in the introduction - it is better suited for the abstract or discussion. therefore I would remove it.

Removed from the Introduction and moved to the discussion.

6. (Methods, Statistics) I think this entire section needs a comprehensive edit and restructuring.

Thanks for the comments on this section, we appreciate the attention paid to the statistical methods in the paper.

a) The models you present in your results section are not adequately described in the methods section. the model fixed and random effects (as described in table 1) should be presented in the method section. you should also provide rationale for the selection of your random effects.

The model used in the analysis has been added to the main body of the text. Further, the

results of the Anova with Satterthwaite's approximation has been added to Table S1 in the supplement. A rationale for the selection of random effects has also been added.

b) Also, It is very uncertain how you arrived at your 'minimum adequate model'. And the accompanying R code does not demonstrate any model comparisons except for the full model and the 'minimum model'. A cursory look at your dataset suggests this is not the most parsimonious model - indeed, quick but uninformed model selection suggests a model without hive "log10(length) ~ body_part + log_mass + (1 | temp/file_name) + body_part:log_mass" is better (based off AIC). However, I understand you are interested in differences between hives so I would revise and think about how best to arrive at the 'best model'.

We have re-tested our models using the step function (in lmerTest, which enables the stats::step function to work with mixed models), which uses AIC score to ascertain the minimum adequate model. Step() and the alternative using Satterthwaite's approximation produced the same results, a significant three-way interaction that we present in the manuscript. We have amended the methods section to clarify what methods we used and why. We avoided using likelihood ratio tests and AIC scores for the reasons outlined in Luke, S. G. *Behav. Res. Methods* 49, 1494–1502 (2017). We appreciate that the previous version of the methods was unclear and thank the reviewer for their suggestions.

c) Is model selection really necessary? Perhaps you could just state your hypothesis - differences between colonies - and describe this.

We believe that this question has been addressed with the above comment in (b).

d) Also, I believe the factor analysis should either be described in more detail or removed as I am uncertain how much it adds in terms of explaining allometric differences between colonies than the mixed effect models. Are there biological reasons to expect the observed clustering? Or is this just random? And what does this tell us about intraspecific allometric scaling? that there is convergence in deviations in intercepts/slopes of allometric relationships. Are the clusters captured by the contrasts derived from mixed models?

The mixed effects models highlight which individual slopes are significantly different from other slopes whereas the FAMD was included to ascertain if some colonies are more similar than others. The clustering tells us that some colonies have slopes dissimilar to all other colonies. The FAMD, rather like a PCA, informs on which factors are most responsible for explaining the variation in our data, for example, colony affiliation is most important for explaining the variability because it has the largest contribution to the first dimension.

e) Also, if part of the aim was to compare allometric slopes between colonies, this can be done using the emtrends function in emmeans.- perhaps this would be a simpler approach to address the same question.

Thanks again for the suggestion, this was indeed how the slopes were compared. The methods (section iv) have been changed to make this clearer.

7. (Discussion, Overall) I find the discussion is too long (8 pages) and overly speculative and i recommend it is considerably edited prior to publication. I do not think speculation is wrong, rather it is the result of allometric differences between observed between colonies when we do not know the reason for them occurring. However, i think the authors should consider which points are most important and discuss these. For example, the two paragraphs about temperature and genetic relatedness among commercial colonies are unnecessary in my opinion.

Thank you for the suggested improvements. Both paragraphs have been removed from the discussion.

Appendix C

The authors investigated organ to body size scaling relationships across bumblebee colonies from commercial sources. They found significant variation in organ scaling between bumblebee colonies and suggest that this may allow for differences in behavior, particularly in spatiotemporal foraging patterns. Although I suggest that the authors approach any evolutionary or adaptive explanations or conclusions based on commercially-bred bumblebees with great caution, I do think that this study poses a significant contribution in the field. It is rare for a study to consider intraspecific differences in morphological scaling, and I believe that this, along with a discussion of its implications, has value. Nonetheless, there are several areas where substantial improvements can be made, detailed below.

General comments:

1) I am concerned with the evolutionary and natural selection-based conclusions that the authors make based on data collected from commercially-bred bumblebees. I agree that the *potential* for variation among and within bumblebee colonies is evident, based on the data presented.

However, because the colonies are commercially reared, this study does not provide any evidence of similar levels of natural variation in the field. Thus, I believe that the claims that natural selection is or has acted on the morphology of these individuals and colonies to generate the observed variation is unfounded. Some supporting data or evidence from field colonies that are more directly subject to natural selection would greatly complement this study.

2) I find some of the key terms and phrases that the authors use frequently throughout the manuscript to be ill-defined or problematic:

- “developmental instability, noise, and/or optimum”: This terminology needs to be clearly defined.

- “allometry” or “allometric” when referring to the field or methodology: I understand that this terminology has often been used in this way. However, the terms “allometric” and “isometric” have mutually exclusive definitions. Despite this, the authors refer to the field of “allometry,” which encompasses both terms/morphological variants. I find distinguishing between the two definitions of “allometry” throughout the manuscript to be challenging. I would suggest replacing “allometry” or “allometric” when referring to the field of study or methodology with “scaling,” or something similar. For example, instead of starting the abstract with the phrase “organ allometry,” which could be incorrectly assumed to mean “disproportionate scaling of organs” by a reader, I would suggest the phrase “organ scaling.”

- “rules”: This terminology is fine when referring to the idea that morphological scaling is subject to certain rules, an idea that this manuscript provides evidence against. However, since this study reveals much individual- and colony-level variation, this terminology is more out of place when referring to the results of the study and the different “rules” that colonies are following. For example, the usage of this terminology is appropriate in lines 410-412, but less appropriate in lines 412-414.

- scaling relationships at “higher levels of organisation” or “broader morphological investigations”: It is unclear what is meant by these phrases or what scale the authors are referring to.

- “slope shifts”: This terminology needs to be clearly defined or should be replaced with something more intuitive – I think the authors simply mean “differences in slope.”

3) I found the writing to be quite vague in some parts of the manuscript. I have identified the most problematic sections in the line-specific comments below.

Abstract:

L18-20: This reads as if you investigated the relationships between each of the three organs, when instead you investigated the morphological scaling of each of the three organs *in relation to body size* and subsequently compared them. Scaling relationships between each of the organs might be an interesting addition to the manuscript, however.

L20-22: “despite the absence of environmental variability and their differing commercial sources”: In a few places in the manuscript, you mention that the colonies were reared at different temperatures. I would not consider this an absence of environmental variability, although I understand your claim that the temperature differences did not account for the morphological variability. Additionally, the idea of finding significant morphological variability *despite* differing commercial sources is confusing – wouldn’t variation based on commercial source be expected?

Introduction:

Overall: For a journal covering a broad range of science topics, this introduction would benefit from more background on social insect biology, particularly regarding colony vs. individual level fitness benefits/costs. For example, more background is needed in the paragraphs from lines 76-100. Background on the importance of worker allometry to task allocation or general colony functioning would also be useful. Many readers may not understand the role that worker variability plays in colony fitness.

L32: “The control of organ size is a central question in animal biology”: You do not appear to address this central question, which I would expect to require investigations into the physiological or developmental control of organ size. Thus, I would suggest removing this part of the sentence and simply begin the sentence with “The relative size of organs...” that you include later in the sentence.

L34-37: I suggest adding “, of higher abundance or nutritional quality.” after “superior resources.” A reference is needed for this idea. You can then split this sentence into two, starting the next sentence with “Therefore, organs in...”

L37-40: I suggest rewording this sentence and perhaps splitting it up for clarity. Something like: “The relationship between organ size and body size can be isometric (organ size scales proportionately with body size) or allometric (organ size scales disproportionately with body size). Organ allometry can be positive, in which organs are relatively larger per unit body size, or negative, in which organs are relatively smaller per unit body size.”

L40-42: I suggest rewording to “Changes in slope and intercept...” and moving this sentence from its current location to after the sentence ending with “differences in slope” on line 47. Also, this sentence is missing a reference, as the authors appear to indicate.

L53: replace “how it presents” with “these relationships”

L54-64: I find this portion of the paragraph very vague. Providing more explanations of these concepts or some concrete examples would be useful.

L65: You refer specifically to morphological variability “within a species” here, but the concepts that you discuss later in the paragraph can be just as easily applied to variability across species and many other levels of biological organization. If there are concepts unique to the intraspecific level, elaborate on them here. Otherwise, I suggest cutting “within a species.”

L67-69: A reference is needed to support this idea.

L73-75: Is this statement referring to only bumblebees, all social insects, or insects in general? If the former, provide a reference to support this claim. If the latter, a reference and/or concrete example would be useful.

L80: replace “body” with “organ”

L81: start a new sentence at “therefore,...”

L84: insert “intraspecific” after “explored” and “social insect” after “among”

L97-98: rephrase to: “... exist in foraging, learning, the degree of worker specialisation, and aggression.”

L98-100: I assume you mean that this phenomenon is unknown in bumblebees. If so, make this clear.

L109-110: replace “how” with “organ scaling among”; remove “scale their organs”

L109-116: Due to the presentation of results and the interpretive nature of this paragraph, I think it would be a great opening paragraph for the discussion!

Methods:

L123-124: I suggest adding information about which colonies were reared at each of the two temperatures, at least in the supplemental materials. Was this based on the commercial source of the colonies?

L124: replace “climate control” with “climate-controlled”

L130: remove “individual”

L135-136: Perhaps describe what equipment you used to photograph the antennae, as you do below for the forewings.

L146: remove “subsequent”

L158-159: An explanation for why you used different magnifications, depending on the colony, is needed.

L161-164: References are needed for these methods.

Results:

- Only 5 out of 12 colonies are included in the within- and across-colony scaling relationship results. More information is needed for the reader to gain a broader understanding of your results: 1) Within colonies, are the colonies or scaling relationships that you don't explicitly mention all isometric or allometric? Are the slopes and intercepts the same for each organ for these unmentioned colonies? 2) Can the reader assume that any across colony comparisons that are not explicitly mentioned have similar/identical scaling relationships? 3) What did the scaling relationships look like depending on temperature treatments or commercial sources? 4) Was mean body size different across colonies, temperature treatments, or commercial sources? This is particularly relevant to the information reported in L225-229. 5) It would be good to report R^2 values for your regression lines to get an idea of the degree of within colony variation. This could be reported as a table in your supplemental materials, with notable variation mentioned in the main text of the results.
- I suggest rearranging the order in which you present your mixed effects models, starting with broader relationships first. For example, it's easier to orient the reader if you present the colony-level scaling data before you then compare this data across colonies. Here is a suggested order of sections, based on your subheadings: 1) organ size independent of organ identity, 2) mean organ size (intercept) across colonies, 3) organ slopes independent of colony affiliation, 4) comparing slopes of organs within colonies, 5) comparing slopes of different organs across colonies.

L211-224: The written explanation of the scaling relationships in this section is less concise than in the previous section. For example, “for a unit increase in body size of bees from colony three, their wings will get proportionally larger than their antennae” can be rewritten as “as body size increases, wing size increases faster than antennae size,” which is more concise and similar to the previous section's writing.

L240-241: replace “negative allometry. Smaller” with “negative allometry, meaning smaller”

Discussion:

L258: The phrase “There exist mechanisms for generating allometric flexibility within a species” is very vague and requires more explanation.

L259: replace “will” with “can”

L262-264: The sentence starting with “We compared...” is redundant with the previous sentence. I suggest you cut it.

L270: The reader will likely not remember the specific differences observed in colony nine. Restate these differences here.

L286-294: This information appears to be restating the results in a similar level of detail. I suggest summarizing and interpreting this information here instead.

L301-307: I was not aware that bumblebees possessed different morphological worker castes, and I don't know of any published data to support this idea. Are you claiming that your data provides evidence for the presence of morphological worker castes in bumblebees? If so, what is the basis of the definition of worker castes you are using here? I would find such a conclusion based on the data presented to be very problematic. Alternatively, do you simply mean that these colonies have different proportions of *differently-sized* workers?

L308-318: Please refer to the first general comment above. I find making conclusions about the adaptiveness of the observed variability in scaling relationships based on commercially-bred bumblebee colonies to be very problematic.

L331-332: replace “which will change if the relationship between organ and body size changes” with “which may change with body size as scaling relationships change”

L350-361: Here, the authors use evidence from two across-species studies to draw conclusions about potential differences in across-colony intraspecific behavior. This should be approached with great caution. There may be other differences across species, aside from variation in organ scaling, that account for spatiotemporal differences in foraging. Additionally, the scale of this variation may be very different in the cited studies than in the current study.

L392-393: The idea that feeding the colonies *ad libitum* removed any variation in nutrition that could account for differences in morphology is flawed, particularly in light of the authors' previous claims that differences in morphology will lead to differences in foraging behavior and resource exploitation.

Table 1: I think this table is well suited to the supplemental material.

Table 3: I suggest reducing the number of significant figures to 2, to be consistent with Table 2.

Figure 1: I think this figure is well suited to the supplemental material.

Figure 2: The colors in this figure are very difficult to distinguish. Perhaps the authors could use a combination of colors, solid vs. dashed lines, and point shapes instead.

Figures 4 & 5: The use of some of the same colors for these figures as those used in Figure 2, in which the colors represent colony identity, is confusing. I suggest using entirely different colors. Alternatively, Figure 4 does not require different colors to be clear, and the authors could use different point shapes to represent clusters in Figure 5.

Appendix D

This manuscript is greatly improved, and I appreciate the authors' efforts to address previous comments. That said, there are a few areas, indicated below, that could use further improvement.

Methods:

Lines 113-115: I think it makes more sense to start the paragraph with this sentence, so the methods read in the same order in which they were carried out.

Thank you for the suggestion, we have moved the lines to the beginning of the methods section.

Results:

Lines 188-192: It took me a few reads to understand the inter-colony relationships here. This section could be revised to be more succinct. For example, assuming my interpretations are correct, something like the following may be more clear: "All colonies had similar worker mean fresh body masses, aside from colonies two and eleven, which tended to have workers of higher mass."

This is not quite the meaning we intended to give, the section in question has been modified. We hope this is clearer in meaning, though it is unfortunately still a little verbose.

Discussion:

Lines 267-274: Most of this still very much reads like an extension or restatement of the results. I think some of this can either be moved up to the results or removed entirely (when redundant with the results).

This section has had some redundant sentences removed, but we feel it is useful to reiterate some parts of the results section for those readers that skim through it or skip it entirely.

Line 287-291: I don't think that you can claim that morphological differences have been selected for in lab-reared bumblebee colonies with long ancestral histories of lab-rearing. The potential for allometric differences among colonies is certainly interesting and may suggest a possibility of selection in a field setting, but I don't think the claim that the findings of this study are the result of selection is appropriate. Perhaps, instead of making that claim here, you can dedicate a paragraph in the discussion to the idea that across-colonies allometric differences open the door for selection to act on colonies that are experiencing different environmental conditions, which would fit well in section ii of your discussion.

We agree that perhaps the statements made regarding the role of selection are a little strong. The paragraph has been modified to reflect the lab-based source of the colonies, with minimal selective pressure, and moved the section ii, as recommended.